# Readiness to prescribe: Using educational design to untie the Gordian Knot

**Ciara Lee**[1☯¤], **Richard McCrory**[1☯], **Mary P. Tully**[2☯], **Angela Carrington**[3☯], **Rosie Donnelly**[4☯], **Tim Dornan**[1☯]*

1 Centre for Medical Education, Queen's University Belfast, Belfast, United Kingdom, 2 Division of Pharmacy and Optometry, University of Manchester, Manchester, United Kingdom, 3 Belfast Health and Social Care Trust, Belfast, United Kingdom, 4 South-Eastern Health and Social Care Trust, Belfast, United Kingdom

☯ These authors contributed equally to this work.
¤ Current address: Department of General Practice and Rural Health, University of Otago, Dunedin, NZ
* T.Dornan@qub.ac.uk

## Abstract

### Introduction

Junior residents routinely prescribe medications for hospitalised patients with only arms-length supervision, which compromises patient safety. A cardinal example is insulin prescribing, which is commonplace, routinely delegated to very junior doctors, difficult, potentially very dangerous, and getting no better. Our aim was to operationalise the concept of 'readiness to prescribe' by validating an instrument to quality-improve residents' workplace prescribing education.

### Methods

Guided by theories of behaviour change, implementation, and error, and by empirical evidence, we developed and refined a mixed-methods 24-item evaluation instrument, and analysed numerical responses from Foundation Trainees (junior residents) in Northern Ireland, UK using principal axis factoring, and conducted a framework analysis of participants' free-text responses.

### Results

255 trainees participated, 54% women and 46% men, 80% of whom were in the second foundation year. The analysis converged on a 4-factor solution explaining 57% of the variance. Participants rated their capability to prescribe higher (79%) than their capability to learn to prescribe (69%; p<0.001) and rated the support to their prescribing education lower still (43%; p<0.001). The findings were similar in men and women, first and second year trainees, and in different hospitals. Free text responses described an unreflective type of learning from experience in which participants tended to 'get by' when faced with complex problems.

**Data Availability Statement:** Quantitative Data: Anonymised quantitative Data is available on the Pure data repository, URL: https://pure.qub.ac.uk/admin/workspace.xhtml?uid=6. Qualitative Data:

Participants in our questionnaire consented to participate in the study, which stated that identifying information would remain confidential. However, they did not provide consent to have their full qualitative responses made publicly available, which would potentially lead to identification of participants and their organisations. Publishing the free-text/qualitative responses beyond the illustrative excerpts included in the manuscript would violate the ethical approval terms from the Queen's University Belfast ethics committee. A committee has been assembled for the purpose of reviewing access to this dataset, the members are Prof Tim Dornan (Queen's University Belfast), Dr Mary Tully (University of Manchester) and Dr Inez Cooke (Queen's University Belfast). If access to the qualitative responses is required, researchers should contact Ms Deborah Millar, Research Administrator, Centre for Medical Education, Queen's University Belfast. Email: Deborah.Millar@qub.ac.uk.

**Funding:** The Research and Development Division of the Public Health Agency of Health and Social Northern Ireland awarded TLD Grant no RES/5199/15 - URL: https://www.publichealth.hscni.net. The funder had no role in study design, data collection and analysis, decision to publish, or preparation of the manuscript.

**Competing interests:** The authors have declared that no competing interests exist.

## Discussion

Operationalising readiness to prescribe as a duality, comprising residents' capability and the fitness of their educational environments, demonstrated room for improvement in both. We offer the instrument to help clinical educators improve the two in tandem.

## Introduction

From the moment they qualify, doctors carry out safety-critical tasks, which can compromise patient safety.[1] Prescribing exemplifies this. Foundation trainees (FTs: junior residents) write 70% of UK hospital prescriptions. The prevalence of errors in these prescriptions is 8% in first-year and 10% in second-year FTs.[1] Patients are not the only ones harmed. FTs are anxious about the possibility of making errors and can be 'second victims' when they make one.[2,3] Prescribing insulin poses a particularly great challenge because it is an everyday task whose rate of errors is even higher than for prescriptions in general.[4] These errors really do harm patients: hypoglycaemia, and/or hyperglycaemia occur on four out of seven hospital-days and one in 25 patients develops ketoacidosis <u>after</u> admission.[4] Given that a patient with diabetes occupies one in six United Kingdom (UK) hospital beds, this is a huge problem.[4] The propensity of new doctors to make costly errors and the toll of errors on both them and patients make learning to prescribe, particularly insulin, an important target for quality improvement.

We should not be surprised that these errors are made because FTs consistently report feeling inadequately prepared to prescribe [3,5–20], most particularly to prescribe insulin.[3] The education provided by medical schools contributes to FTs' unpreparedness. Curricula and assessments focus mainly on acquiring knowledge and skills 'off-the-job',[14] which does not fully emulate the 'on-the-job' complexity of prescribing that leads to errors.[1,21] This type of education led, in one study, to more graduates rating themselves ready to write prescriptions than to prescribe safely.[22] Induction programmes and assistantships make the transition from off-the-job learning in medical school to on-the-job learning after qualification less stressful [2,3], particularly when students are supported by pharmacists,[19] but evidence that assistantships improve prescribing safety is lacking.[16]

The net result is that new graduates undergo an abrupt transition from medical students, highly supervised and, in many countries, protected from workplace pressures, to FTs performing a key safety-critical task whilst fully exposed to workplace pressures under sometimes limited supervision. Confidence rises quickly so that, within just 8 months of qualifications, 60% of FTs report feeling confident to prescribe insulin independently.[23] Their error rate, cited above, shows that their confidence is not always matched by their competence. FTs have been reported to use guesswork rather than knowledge and skills learned in medical school to guide prescribing and do not always check the accuracy of their prescriptions.[15] This conforms to a wry definition of clinical experience as 'making the same mistakes with increasing confidence over an impressive number of years'.[24]

Current literature, in sum, shows limitations in the ability of undergraduate education to prepare new graduates to prescribe and to lay down patterns of safe practice when education shifts to practice-based learning. It shows a dearth of pedagogic research and development to guide learning during this all-important phase in FTs' professional development. In order to advance the field, researchers have developed two instruments purporting to measure how well learners are prepared to prescribe: one for FTs;[8] and one for medical students.[20] One

was psychometrically validated [20] but the other was not.[8] Both evaluate what students were taught but not what they learned. They do not have clear theoretical underpinnings and treat preparedness as a final outcome rather than a process that continues throughout doctors' working lives. This creates a pressing need to theorise and operationalise readiness to prescribe.[16]

Scholars tell us that members of all trades and professions are readied to work by engaging in work activities, observing attentively, gaining supervised experience of the physical and social settings where work is done and, ultimately, being productive members of workforces. [25,26] They learn by participating in the talk of practice.[27] Learning is not a one-way process [19,28] because, as Lave and Wenger argued,[29] learners' participation changes communities of practice as well as the reverse. These theoretical insights define readiness for practice as an attribute that cannot be fully assured by off-the-job education alone. It seems timely to examine how prescribing education can benefit from a conceptualisation, which acknowledges the intimate relationship between individual readiness and the readiness of work environments to support on-the-job learning.

Regehr urged education researchers to 'represent complexity well'.[30] He advocated responding to the 'imperative of understanding' as opposed to the 'imperative of proof'.[30] Shedding light on complex educational systems requires novel methodologies, of which design-based research (DBR) is one.[31] 'Designs' are programmatic approaches to complex problems, which researchers apply iteratively, evaluating their effects, progressively refining the designs, and thereby understanding and gaining traction on the problems. DBR harnesses the 'observer effect' according to which observing a phenomenon changes that phenomenon. It does not force dichotomies between observation and intervention, or between theory and practice. We chose insulin therapy as a cardinal example of unreadiness to prescribe and launched the first iteration of a design-based research programme to address it. We framed the research question: How can readiness to prescribe be conceptualised and operationalised? To answer the question, we designed, implemented, and evaluated a mixed methods investigative instrument.

## Methods

### Research ethics approval

The Joint Research Ethics Committee of the School of Medicine, Dentistry, and Biomedical Sciences of Queen's University Belfast approved the project (approval no 17.33v3). The Committee judged that the project had limited potential to cause harm and did not require written consent because no personal details were recorded. The questionnaire had a cover page, which explained participants' right not to complete it but invited them to give implicit consent by doing so.

### Theoretical orientation

Since the goal of our programme of DBR was to understand and improve FTs' clinical behaviour, we chose a leading behaviour change theory to guide the design of our investigative instrument: Michie and colleagues' Capability-Opportunity-Motivation-Behaviour (COM-B) theory.[32] According to this, learners adopt desired behaviour when motivated to do so. There are two types of motivation: reflective (conscious) and automatic motivation (habit). Learners' motivation is influenced by their psychological capability and the social opportunities their practice environments provide. Learners' increasing capability exercises positive feedback on their learning environments by helping them get the best out of their fellow workers and available educational resources. COM-B is compatible with our theoretical orientation towards complexity because it regards behaviour change as an open system in which

conditions and processes feed back on one another through a web of actions and interactions. Since patient safety depends on this, we regarded it as axiomatic that capability should mean prescribing when psychologically capable to do so independently and calling for supervisory help when not. Since receiving help on earlier occasions makes trainees more psychologically capable to prescribe independently on subsequent occasions, readiness is a shift towards increasingly independent behaviour rather than an endpoint.

Because our goal was to operationalise the concept of readiness, we drew insights from another theoretical framework that explains complex relationships between contexts and processes, the Consolidated Framework for Implementation Research (CIFR).[33] This provides empirically grounded constructs that predict the success of interventions and complement COM-B: reflective motivation in COM-B, for example, corresponds to self-efficacy in CIFR. We transcribed relevant constructs from COM-B and CIFR into an Excel (Microsoft, Redmond USA) spreadsheet and cross-mapped them. In addition, we triangulated our interpretation of these theories against Reason's theory of error causation [34] and empirical research into the causes of FDs' prescribing errors [21,35]. The members of our team, which comprised a medicines governance pharmacist, a diabetes specialist pharmacist, a junior doctor, a senior doctor, and a lay member (administrator) reflexively clustered the items and progressively reduced them to a comprehensive set of items without obvious redundancy. In summary, we conceptualised readiness as a multi-dimensional construct representing learners' capability and motivation to prescribe within the constraints and opportunities provided by the contexts in which they learn to practise and/or to recruit additional resources in order to assure patient safety and learn by doing so.

## Setting

The project was conducted in Northern Ireland, a geographically bounded province of the UK, which has a population of 2 million people. Healthcare is provided by five Health and Social Care Trusts, in which 500 FTs are educated. The 2-year Foundation Programme is managed by a single Training Authority. Trainees complete 4–6 month hospital placements, including medicine, surgery, and a range of other secondary and tertiary care specialties. Some second-year FTs also gain experience in general practice.

## Study design, recruitment, and participants

The study had a survey design and an opportunistic sampling strategy. All FTs were eligible to participate without any exclusion criteria. The goal was to recruit as many of them as possible from all five Trusts over a 12-month period. Recruitment was low despite team members assiduously attending locality teaching sessions because FTs attended teaching poorly. In light of this, we recruited at regional quality improvement teaching, which is mandatory, and attended by second-year FTs (FT2s) from all Trusts. A team member briefly explained the study during a drinks break or at the end of the session and non-coercively invited FTs to complete the questionnaire. She did not stand over them as they completed it, and the form required no personal details. The teaching sessions were not directly related to insulin safety. The recruitment process led, inescapably, to over-representation of FT2s in the sample. The ratio of the number of FT1s to the number of items (2.1:1) was too small for FT1s to be analysed as an independent sample, so we pooled all respondents for statistical analysis.

## Instrument

The spreadsheet containing cross-mapped constructs from the underpinning theories provided items for the questionnaire, which we further refined during several further phases of

piloting and revision. The questionnaire asked participants to rate their agreement with 24 short statements using anchored 7-point Likert scales. It also contained comment boxes, inviting participants to give free-text reasons for their numerical ratings. Guided by analysis of pilot data, we organised the numerical items into five clusters and placed a comment field after each of these. The instrument as used here (it has been refined subsequently in response to these findings–see Discussion) is included as S1 Fig.

## Statistical analysis

**Data cleaning and removal of multivariate outliers.** We excluded the responses of four participants who answered fewer than half the items. Three percent of the remaining scores were missing, which we imputed using the Expectation Maximisation algorithm in SPSS v25. This changed the mean value of only one item by 1%, which suggested imputation had not significantly distorted the findings. The Mahalanobis Distance technique at a critical alpha value of 0.001 [36] identified seven questionnaires with multivariate outliers. Those two stages of data cleaning reduced the dataset from 266 to 255 participants.

**Exploratory factor analysis.** The Kaiser-Meyer-Olkin measure of sampling adequacy (0.75) and Bartlett's test ($p<0.001$) suggested a factorable solution was possible for the 24 items. Since the items were theoretically linked and statistically inter-correlated (r values varying between 0.02–0.71 for bivariate correlations) analysis was by Principal Axis Factoring with oblique (Direct Oblimin) rotation. The scree plot showed an inflexion point below the fourth factor so we examined possible solutions around this factor value. For interpretive purposes, we considered an item relevant to a specific factor if the absolute value of the standardized loading was greater than 0.3, and the factor loading was at least 0.2 higher than other loadings. A four-factor solution accounted for the highest estimated variance in the sample and produced more substantial loadings for each variable than other solutions. We used Cronbach's Alpha, inter-item and item-total correlations to examine the internal consistency of these subscales. There were three items whose alpha-if-deleted coefficients were greater than the alpha coefficient for the factor they loaded onto, which we deleted before computing the properties of a final 20-item version, whose items are shown in Table 1.

**Descriptive and correlation analysis.** The analysis had three final stages: 1) Calculating an aggregate mean of the items loading to each factor for each participant (expressed as percentage of the scale maximum), computing group grand means, and testing for differences between them using a Kruskal-Wallis test. The instrument includes several negatively-worded items, whose valence we reversed by subtracting the median and quartile values from 100. 2) Examining the relationship between those factors and year of training (FY1 or FY2) and sex. 3) Selecting the four hospitals with over 30 participants attached to them and analysing relationships between hospital and factor scores.

## Qualitative analysis

One hundred and seventy-eight participants gave free-text responses to at least one item. Having entered these into a spreadsheet, the first author and senior author read them, jointly devised, and then applied a simple coding framework informed by our theorisation of readiness to prescribe. This included attributes of: FTs as individuals; their job; other individuals with whom they worked or were supervised; the culture in which they worked; and the tools that were available to support their work. Responses could code to more than one of those categories. The first author wrote a detailed precis of the contents of the table. She and the senior author then condensed the data across categories and domains.

**Table 1.  Results of principal axis factoring.**

| | Factor loading | Median (IQR) | Alpha if deleted |
|---|---|---|---|
| **Factor 1: Capability to learn—Cronbach's alpha = 0.73** | | | |
| I am in the habit of consulting books/online resources/guidelines to help me prescribe | 0.71 | 3 (2–5) | 0.67 |
| I am in the habit of discussing prescriptions with other doctors (seniors or peers) | 0.70 | 5 (4–5) | 0.70 |
| I use learning tools to increase my knowledge and skills | 0.63 | 4 (3–5) | 0.67 |
| When I am unsure what is the right action, I seek guidance | 0.60 | 5 (5–6) | 0.70 |
| I am in the habit of discussing prescriptions with nurses or pharmacists | 0.57 | 4 (3–5) | 0.73 |
| I (would) like to receive constructively critical feedback on my prescriptions | 0.57 | 5 (4–6) | 0.72 |
| **Capability to learn (% of scale maximum)** | | 69% (61–81) | |
| **Factor 2: Capability to prescribe—Cronbach's alpha = 0.81** | | | |
| I am confident I am on the path to being a good prescriber | 0.78 | 5 (4–6) | 0.77 |
| I feel safe to put into practice what I learn about prescribing | 0.74 | 5 (4–5) | 0.77 |
| I can distinguish simple prescribing decisions from difficult/ambiguous ones | 0.66 | 5 (4–5) | 0.80 |
| When I recognise what action needs to be taken, I prescribe without hesitation | 0.65 | 5 (4–5) | 0.81 |
| I can judge whether my knowledge and skills are sufficient for individual prescribing decision | 0.65 | 4 (4–5) | 0.80 |
| I expect my foundation education will result in me prescribing well | 0.61 | 5 (4–5) | 0.80 |
| I think out prescriptions logically rather than by habit | 0.60 | 5 (4–6) | 0.79 |
| **Capability to prescribe (% of scale maximum)** | | 79% (71–86) | |
| **Factor 3: Tensions–Cronbach's alpha = 0.83** | | | |
| Tensions with other health professionals (e.g. nurses/pharmacists) affect my capability to prescribe well | 0.93 | 2 (1–4) | Not applicable as only 2 items |
| Tensions with senior or junior doctors affect my capability to prescribe well | 0.89 | 1 (1–3) | |
| **Tensions (% of scale maximum)** | | 33% (17–50) | |
| **Factor 4: Support—Cronbach's alpha = 0.87** | | | |
| The people where I work give me constructively critical feedback on my prescribing | 0.91 | 2 (1–3) | 0.81 |
| The people where I work give credit for good prescribing | 0.87 | 2 (1–3) | 0.84 |
| The people where I work encourage/support me to reflect critically on the quality of my prescriptions | 0.80 | 2 (1–4) | 0.84 |
| The people where I work make a virtue out of acknowledging uncertainty and seeking help | 0.73 | 3 (2–4) | 0.88 |
| The people where I work support my learning to prescribe | 0.69 | 4 (3–5) | 0.86 |
| **Support (% of scale maximum)** | | 43% (27–60) | |

# Results

Two hundred and fifty-five FTs participated (54% women and 46% men; 80% FT2s and 20% FT1s). Participants worked in thirteen hospitals and several general practices, representing all five Trusts.

## Principal axis factoring and quantitative comparisons (Table 1)

The analysis converged in 11 iterations on a 4-factor solution, explaining 57% of the variance in the data. Participants rated their ability to prescribe higher (79%) than their ability to learn to prescribe (69%; p<0.001). They rated the support to their prescribing education rather low (43%; p<0.001 compared with their capability to prescribe or their capability to learn).

The one negatively worded item—Tensions—was also low (33%), suggesting that tensions with doctors or other health professionals were not a main influence on participants' prescribing education. We found no significant differences between the factor scales when comparing male and female participants, FT1s and FT2s, and hospitals.

The qualitative findings are summarised in Table 2.

**Table 2. Results of qualitative analysis.**

| | What increased FTs' capabilities to (learn to) prescribe? | What reduced FTs' capabilities to (learn to) prescribe | Missed opportunities (what was absent in participants' learning environments) |
|---|---|---|---|
| **FDs themselves** | • Practice<br>• Experience<br>• Good understanding of insulin types | | • Reflection on prescribing |
| | *Exemplar quotations*:<br>'With experience I'll get better' | | |
| **Community** | • Advice from senior doctors, DSNs, and occasionally nurses and pharmacists<br>• Constructive verbal feedback<br>• Teaching<br>• Pharmacists picking up errors<br>• Following the prescriptions of others<br>• Good documentation of management plans | • Non-constructive criticism<br>• Poor communication<br>• Tensions with nurses<br>• Following the prescriptions of others | • Insufficient feedback on prescribing<br>• Teaching<br>• Credit for good prescribing<br>• Encouragement to reflect on prescribing<br>• Missed opportunity to engage with some patients<br>• Use of other FTs as a resource |
| | *Exemplar quotations*:<br>'DSNs very approachable—good to discuss issues with'; 'love chatting to them, incredibly helpful'<br>'Pharmacists picking up prescribing errors', and 'checking and flagging up any mistakes I make'<br>'if patient is actively involved, better education and ownership', 'They know their own bodies' (Teaching could help) 'to understand the different types of insulin and insulin regimes better' | *Exemplar quotations*:<br>'Nurses bring the kardex to you rather than you to the patient'.<br>'Nurses often hand you kardexes and rush you to prescribe', 'Nurses telling me to do opposite thing'.<br>'No well documented plan',<br>'No clear documentation from DSN on dose adjustments.' they 'just changed the prescription without telling me what was wrong';<br>'Seniors disagree with my clinical judgement without explaining their clinical reasoning/rationale'.<br>'If someone else before me has prescribed incorrectly and I copy what they did . . .' | *Exemplar quotations*:<br>'never get any feedback'<br>'very rarely get feedback,<br>'there is no feedback'.<br>'. . .you don't get any feedback whether you have done the right thing the next morning'.<br>'no credit is given for good prescribing'.<br>'No real follow up when insulin prescribed' |
| **Environment** | • Supportive learning environments | • Lack of access to advice and support (especially out-ot-hours)<br>• Distractions<br>• Prescribing away from the bedside<br>• Systemic hypophobia<br>• Unfamiliarity with patients (covering wards/shifts) | • Support systems / availability of advice out-of-hours |
| | | *Exemplar quotations*<br>'unnecessary pressure to prescribe quickly' and 'time pressure' whilst responding to 'high workloads'.<br>'Distractions on the ward' and 'interruptions while prescribing'<br>'unwillingness from nursing colleagues to administer insulins in lower BMs' and 'Nursing staff frightened of hypos and would rather omit insulin'. | |
| **Tools and Guidelines** | • Well-designed prescription charts | • Difficulty finding and accessing guidelines | |
| | *Exemplar quotations*<br>'XXX (see table footnote) Trust has easy-to-use insulin chart with the different types of insulin outlined on the back—I feel safer'. | *Exemplar quotations*<br>Knowing 'where/how to access practically useful information' and 'where the resources are' and having 'access to guidelines". | |
| **The job of an FT** | | • Workload<br>• Time pressures<br>• The inherent complexity and uncertainty of prescribing insulin<br>• Prescribing in difficult/complex scenarios | |
| | | *Exemplar quotations*<br>'More complicated patients', 'difficult prescriptions' and 'variability in each individual situation'<br>'Prescribing tends to be less straightforward than initially expected—lots of confounding factors'. | |

'XXX' is used to anonymise the Trust for confidentiality reasons

## What made FDs more capable

**People and practice communities.**   Supportive learning environments increased capability. Whilst the support of senior and specialist doctors was important, the expertise and supportive behaviour of diabetes specialist nurses (DSNs) was particularly valued. Their support was, however, least available out of hours when it was most needed and not always available even within working hours. Ward pharmacists were more a safety net than someone to consult with prior to prescribing. Participants sometimes valued patients' involvement in shared decision-making, although confused and very unwell patients could not do this. Feedback was most helpful when it was constructive rather than critical, face to face, and directly related to individual prescriptions.

**Experience.**   Many participants said experience would make them more capable though they did not specify how it would do so.

**Teaching.**   Comments about teaching were equally unclear other than that participants needed help to develop a good working knowledge of different insulin types.

**Tools.**   Well-designed prescription charts and clear documentation of patients' usual doses of insulin and changes to management plans increased capability. Whilst local protocols and guidelines increased capability, these were not always available when needed.

## What made FDs less capable

**Difficult clinical problems.**   Clinical complexity made FDs less capable. There was an obvious contradiction between participants' stated belief that experience would make them more capable, and their statements that difficult situations made them less capable because they were not well supported in managing these.

**Being busy and under pressure.**   Working in busy learning environments with heavy workloads, being pressured to prescribe quickly, frequent distractions, and insufficient time to engage with patients reduced participants' capability. Being expected to prescribe away from patients' bedsides made this worse and relationships with nurses became strained, particularly when nurses feared that prescribed doses would make patients hypoglycaemic. Poor documentation of patients' insulin doses and management plans further reduced participants' capability.

**Unhelpful criticism.**   Unconstructive or frankly destructive criticism reduced participants' learning. Participants were unsure how to respond to this and were left with doubts about their own clinical judgement. They found it unhelpful when other prescribers changed their prescriptions without explaining why.

**Copying what others had done.**   Participants resolved their uncertainty about what to prescribe by copying earlier prescriptions written by peers. They were aware that this might perpetuate poor practice but, in the absence of expert supervision to hand, had nothing better to guide them.

## What was missing

**Feedback.**   Participants would have appreciated constructive feedback but rarely received this: Feedback was least available for out-of-hours work, which left participants wondering if they were doing 'the right thing'.

**Other factors.**   Praise was missing. Heavy workloads and time pressures prevented participants knowing what happened to patients they had prescribed for. Encouragement to reflect on prescribing was rare and usually only followed adverse events. Even when educational resources were available, time constraints limited their use. Participants might have made greater use of patients' expertise. They chose only to discuss insulin prescriptions with 'well

informed', 'competent' patients, 'who normally look after their own regimen and are confident in doing so'. Some comments verged on dismissing patients' roles in their own care, rather than having conversations with patients from which they might have learned: 'I ask sensible patient what they want prescribed'; 'Many patients don't understand their insulin'.

## Discussion

The design of our investigative instrument was guided by conceptualising readiness to prescribe as an interaction between FTs and their learning environments, which leads them to prescribe when capable and recruit additional resources when not capable. The responses of 255 FTs provided validity evidence for the conceptualisation and preliminary evidence about participants' prescribing education. They rated their readiness to prescribe higher than their readiness to learn to prescribe. They identified shortcomings in their learning environments–notably, a lack of support to critical reflection. Participants' free text responses described an unreflective type of learning from experience in which they uncritically copied what others had done before and learned to 'get by' when faced with complex problems unsupported. Workload pressures, for example being presented with several prescription charts away from the bedside and being expected to prescribe quickly without assessing patients, coupled with pressure not to make patients hypoglycaemic, may have encouraged unreflective behaviour.

The findings are important, given that the first two years of practice play a crucial role in doctors' prescribing education. The findings show that our novel instrument meets several validity criteria.[37] Its items were derived from robust conceptual frameworks and bear a logical relationship to the domain being measured. The response process left relatively few numerical items missing, which could be compensated for by imputation. The internal structure had acceptable reliability. We have not yet shown a relationship between our measure and other variables and cannot yet say what impact it will have, though using best available theory to guide its design increases the likelihood that it will have consequential validity. The inclusion of free text items also contributes to the consequential validity of the instrument by identifying ways of improving learning environments, as well as measuring their quality. The quantitative findings identified possibilities for improvement, most important of which is to foster a positive educational culture that values good prescribing, encourages constructive feedback and learning, and promotes greater collaboration with fellow patients and professionals.

### Limitations

One limitation was the relative under-representation of FT1s, which prevented us exploring differences between them and FT2s. Our earlier research, however, suggests that the causes of error are very similar in FT1s and FT2s.[1] The lack of difference between hospitals may suggest the instrument is insensitive. An alternative explanation is the relative homogeneity of the research setting, within a single UK region served by a single deanery and single medical school. These may not, however, represent foundation trainees more widely. Another limitation was using relatively fragmentary qualitative data, rather than in-depth analysis of interviews or focus groups.

### Implications

The main implication is to healthcare quality improvement. The instrument addresses an important problem, has been rigorously validated in the contexts where it will be applied, and is therefore fit for wider implementation and evaluation. It has already identified ways of improving foundation trainees' practice and learning to provide safe and effective insulin

therapy, which could have significant impact on patient safety. Since measuring the status quo, of itself, tends to change the status quo, we suggest the instrument should be used to audit FTs' readiness to prescribe insulin in further DBR cycles. To broaden the applicability of this research, we are now evaluating a 20-item version of the instrument that is not specific to insulin and can evaluate the readiness of nurses and pharmacists, as well as doctors, to prescribe.

## Conclusion

We conclude, pending further research, that there is value in conceptualising readiness to prescribe as a complex construct, which comprises the attributes of individuals, features of their learning environments, and interactions between the two. We have provided validity evidence for a set of numerical items that explore interrelationships between these factors, and free text items that provide information to guide improvement efforts. This research suggests that the Gordian knot of prescribing error might be untied by paying at least as much attention to the social and material environments in which FTs learn to prescribe as on training them individually in prescribing skills. Our findings show room for improvement in the prescribing cultures in which FTs learn and a need to encourage a more reflective approach to foundation education.

## Supporting information

**S1 Fig. Shows the instrument.**
(PDF)

## Acknowledgments

We thank the participants for their time and thoughtful answers.

## Author Contributions

**Conceptualization:** Ciara Lee, Mary P. Tully, Tim Dornan.

**Data curation:** Ciara Lee, Tim Dornan.

**Formal analysis:** Ciara Lee, Richard McCrory, Tim Dornan.

**Investigation:** Angela Carrington, Rosie Donnelly.

**Methodology:** Mary P. Tully.

**Writing – original draft:** Ciara Lee, Mary P. Tully, Rosie Donnelly, Tim Dornan.

**Writing – review & editing:** Ciara Lee, Richard McCrory, Mary P. Tully, Angela Carrington, Tim Dornan.

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
