## [Decision Letter · Decision Letter 0]

8 Oct 2019

PONE-D-19-20466

Readiness to prescribe: using educational design to untie the Gordian Knot

PLOS ONE

Dear Professor Dornan,

Thank you for submitting your manuscript to PLOS ONE. After careful consideration, we feel that it has merit but does not fully meet PLOS ONE’s publication criteria as it currently stands. Therefore, we invite you to submit a revised version of the manuscript that addresses the points raised during the review process.

Please amend your manuscript in light of the reviewer's comments.

All changes to be made to the manuscript should be in Red or Blue colour and underlined, in the revised manuscript with track changes.

Please also submit a response letter, addressing in a point-by-point format each point raised.

We would appreciate receiving your revised manuscript by Nov 22 2019 11:59PM. To enhance the reproducibility of your results, we recommend that if applicable you deposit your laboratory protocols in protocols.io, where a protocol can be assigned its own identifier (DOI) such that it can be cited independently in the future. For instructions see: http://journals.plos.org/plosone/s/submission-guidelines#loc-laboratory-protocols

We look forward to receiving your revised manuscript.

Kind regards,

Samy A Azer, M.D., Ph.D., (USyd), M.Ed. (UNSW), M.P.H. (UNSW),

Academic Editor

PLOS ONE

1. Please include your tables as part of your main manuscript and remove the individual files. Please note that supplementary tables (should remain/ be uploaded) as separate "supporting information" files

Reviewers' comments:

Reviewer's Responses to Questions

**Comments to the Author**

1. Is the manuscript technically sound, and do the data support the conclusions?

Reviewer #1: Yes

2. Has the statistical analysis been performed appropriately and rigorously? 

Reviewer #1: Yes

3. Have the authors made all data underlying the findings in their manuscript fully available?

Reviewer #1: No

4. Is the manuscript presented in an intelligible fashion and written in standard English?

Reviewer #1: Yes

5. Review Comments to the Author

Reviewer #1: Thank you for your submission to the journal. The writing throughout your paper is clear and concise. You have provided an answerable research question and the logic for the choice of question was well explained. The explanation of the methods chosen and the reasons for these was comprehensive and logical. There was a good explanation of the significance of the findings and of the plans for future research.

I have minor questions and suggestions:

Was the population who completed the survey representative of the total population of foundational trainees?

Line 64 please remove the underlining

Line 90 this sentence begins with a conjunction (and) and would read better if restructured.

Line 174 is (2.1) meant to be 2 to 1?

Line 181 re Likert scores - what were the number of response options offered for each question?

Line 234 Nurses and pharmacists are not generally considered as allied health professionals. I suggest the term other health professionals as per the question in the survey.

6. PLOS authors have the option to publish the peer review history of their article (what does this mean?). If published, this will include your full peer review and any attached files.

Reviewer #1: Yes: Kirsten Small

---

## [Decision Letter · Decision Letter 1]

2 Jan 2020

Readiness to prescribe: using educational design to untie the Gordian Knot

PONE-D-19-20466R1

Dear Dr. Dornan,

We are pleased to inform you that your manuscript has been judged scientifically suitable for publication and will be formally accepted for publication once it complies with all outstanding technical requirements.

With kind regards,

Wen-Jun Tu

Academic Editor

PLOS ONE

Additional Editor Comments (optional):

Reviewers' comments:

Reviewer's Responses to Questions

**Comments to the Author**

1. If the authors have adequately addressed your comments raised in a previous round of review and you feel that this manuscript is now acceptable for publication, you may indicate that here to bypass the “Comments to the Author” section, enter your conflict of interest statement in the “Confidential to Editor” section, and submit your "Accept" recommendation.

Reviewer #1: All comments have been addressed

2. Is the manuscript technically sound, and do the data support the conclusions?

Reviewer #1: Yes

3. Has the statistical analysis been performed appropriately and rigorously? 

Reviewer #1: Yes

4. Have the authors made all data underlying the findings in their manuscript fully available?

Reviewer #1: No

5. Is the manuscript presented in an intelligible fashion and written in standard English?

Reviewer #1: Yes

6. Review Comments to the Author

Reviewer #1: Thank you for attending to the requested changes. You are to be commended on the quality of the research and writing.

7. PLOS authors have the option to publish the peer review history of their article (what does this mean?). If published, this will include your full peer review and any attached files.

Reviewer #1: Yes: Kirsten Small

---

## [Editor Report · Acceptance letter]

14 Jan 2020

PONE-D-19-20466R1 

Readiness to prescribe: using educational design to untie the Gordian Knot 

Dear Dr. Dornan:

I am pleased to inform you that your manuscript has been deemed suitable for publication in PLOS ONE. Congratulations! Your manuscript is now with our production department. 

With kind regards,

on behalf of

Dr. Wen-Jun Tu 

Academic Editor

PLOS ONE